# Genome Sequence and Assessment of Safety and Potential Probiotic Traits of *Lactobacillus johnsonii* CNCM I-4884

**DOI:** 10.3390/microorganisms10020273

**Published:** 2022-01-25

**Authors:** Anne-Sophie Boucard, Isabelle Florent, Bruno Polack, Philippe Langella, Luis G. Bermúdez-Humarán

**Affiliations:** 1INRAE, AgroParisTech, Micalis Institute, Université Paris-Saclay, 78350 Jouy-en-Josas, France; Anne-Sophie.Boucard@inrae.fr (A.-S.B.); philippe.langella@inrae.fr (P.L.); 2UMR 7245, Muséum National d’Histoire Naturelle, Centre National de la Recherche Scientifique, Sorbonne Universités, 75005 Paris, France; isabelle.florent@mnhn.fr; 3Anses, INRAE, Ecole Nationale Vétérinaire d’Alfort, UMR BIPAR, Laboratoire de Santé Animale, 94700 Maisons-Alfort, France; bruno.polack@vet-alfort.fr

**Keywords:** probiotic, *Lactobacillus johnsonii*, genome sequencing, identification, safety

## Abstract

The probiotic strain *Lactobacillus johnsonii* CNCM I-4884 exhibits anti-*Giardia* activity in vitro and in vivo in a murine model of giardiasis. The aim of this study was the identification and characterization of the probiotic potential of *L. johnsonii* CNCM I-4884, as well as its safety assessment. This strain was originally classified as *Lactobacillus gasseri* based on 16S gene sequence analysis. Whole genome sequencing led to a reclassification as *L. johnsonii*. A genome-wide search for biosynthetic pathways revealed a high degree of auxotrophy, balanced by large transport and catabolic systems. The strain also exhibits tolerance to low pH and bile salts and shows strong bile salt hydrolase (BSH) activity. Sequencing results revealed the absence of antimicrobial resistance genes and other virulence factors. Phenotypic tests confirm that the strain is susceptible to a panel of 8 antibiotics of both human and animal relevance. Altogether, the in silico and in vitro results confirm that *L. johnsonii* CNCM I-4884 is well adapted to the gastrointestinal environment and could be safely used in probiotic formulations.

## 1. Introduction

Probiotics are defined as live microorganisms that, when administered in adequate amounts, confer a health benefit on the host [1]. Some of the better known beneficial effects of probiotics include stimulation of innate and adaptive immune response, modulation of the host metabolism (through enzymes and metabolites), anti-pathogenic activity, and stimulation of the barrier effect of the intestinal epithelium [2]. Many studies have demonstrated the beneficial effects of probiotics in a wide range of conditions, such as irritable bowel syndrome (IBS) [3], cancer [4], food allergy [5], metabolic syndrome [6], inflammatory bowel disease (IBD) [7], depression [8], *Clostridioides difficile* infection [9] and *Giardia* infection [10], among others. Importantly, the efficacy of probiotics has been shown to be strain- and disease-specific [11]. Probiotics belong mainly to microorganisms of the genera *Lactobacillus*, *Bifidobacterium* and *Saccharomyces*. Lactobacilli have a long history of safe use in traditional fermented foods and beverages, and some species and strains are considered important members of the healthy gut microbiota [12,13]. In addition, many members of the genus *Lactobacillus* are Generally Recognized As Safe (GRAS, Food and Drug Administration, USA), or included in the Qualified Presumption of Safety list (QPS, European Food Safety Authority, EU), which guarantees its use in food and safety in humans, respectively.

*Lactobacillus johnsonii* is a commensal bacterial species present in the gastrointestinal tract (GIT) of many mammals including mice [14], dogs [15], calves [16], yaks [17], pigs [18] and humans [19] associated with gut health. Some strains of *L. johnsonii* have been isolated from traditional fermented products [20,21]. Interestingly, a recommendation of the EFSA panel on Biological Hazards (BIOHAZ) has included *L. johnsonii* in the updated QPS list [22]. *L. johnsonii* has a GRAS status and is therefore considered safe; moreover, several *L. johnsonii* strains have been successfully administered as a probiotic treatment in mice [23,24], rats [25], pigs [26,27], dogs [28] and humans [29,30]. Finally, *L. johnsonii* CNCM I-4884 has been identified as a new probiotic candidate with in vitro and in vivo anti-parasitic activity in a murine model of giardiasis [31].

EFSA has developed guidelines for the safety assessment of probiotics, which include a taxonomic identification of the strain by whole genome sequencing, genomic and phenotypic determination of the antibiotic resistance profile and safety assessment [32]. Furthermore, the host-associated stress resistance and epithelium-binding capacity of the strain should be assessed to ensure that the candidate probiotic can withstand the stressful conditions of the host GIT and transiently colonize the host to exert its functional properties [33].

The aim of this study is to investigate the safety profile and probiotic potential of the strain *L. johnsonii* CNCM I-4884, a Gram-positive bacterium registered in the National Collection of Microorganisms Cultures (CNCM, Pasteur Institute, Paris, France) as a class 1 bacterium. Whole genome sequencing was performed to determine taxonomical affiliation, and in vitro approaches were used to characterize the fermentation profile, resistance to acid and bile salts, and antibiotic susceptibility of the strain.

## 2. Materials and Methods

### 2.1. Bacterial Strain

*Lactobacillus johnsonii* CNCM I-4884 was isolated from a human sample [31]. The strain was stored at –80 °C in Man Rogosa Sharpe (MRS) broth (Difco, Le Pont de Claix, France) plus 20% glycerol until further analysis. The strain was routinely grown in MRS broth at 37 °C under microaerobic conditions without agitation.

### 2.2. Genome Sequencing

Genomic DNA from CNCM I-4884 was purified using the Wizard Genomic DNA Purification kit (Promega, Madison, WI, USA) from 2 mL of overnight culture. The quality of the recovered DNA was confirmed by agarose gel electrophoresis (1%) and both concentration and purity were assessed using a Nanodrop 1000 apparatus (Ozyme, Saint Quentin, France). Genomic DNA was then sequenced using Illumina MiSeq technology (2 × 150 bp; Eurofins Genomics, Constance, Germany). Genome assembly was performed using SPAdes and annotated with the PATRIC RASTtk-enabled Genome Annotation Service [34,35]. The CG View (Circular Genome Viewer) server Beta was used to construct a circular graphical map of the genome. 

### 2.3. Phylogenetic Analyses

The complete genome sequence of the species belonging to the *Lactobacillus acidophilus* group [36] were searched in the NCBI (National Center for Biotechnology Information) database and 18 genomes were selected: *L. johnsonii* UMNLJ21 (GenBank ID: CP021703), *L. johnsonii* UMNLJ22 (CP021704), *L. johnsonii* FI9785 (FN298497), *L. johnsonii* ATCC 33200 (ACGR00000000), *L. johnsonii* DPC 6026 (CP002464), *L. johnsonii* BS15 (CP016400), *L. johnsonii* NCC 533 (AE017198), *L. johnsonii* N6.2 (CP006811), *L. taiwanensis* DSM 21401 (AYZG01000000), *L. taiwanensis* CLG01 (CP059276), *L. gasseri* DSM 14869 (CP006803), *L. gasseri* ATCC 33323 (CP000413), *L. paragasseri* JCM 5343 (AP018549), *L. acidophilus* 20079 (CP020620), *L. acidophilus* La-14 (CP005926), *L. acidophilus* NCFM (CP000033) and *L. acidophilus* LA1 (CP017062). The phylogenetic position of CNCM I-4884 was determined using the PATRIC Phylogenetic Tree Service [37]. For protein comparison analysis, the *L. johnsonii* genomes were translated and compared to the proteins observed in the CNCM I-4884 genome. The comparison and visualization of the percentage identity of the proteins was generated with THE SEED server version 2.0 [38].

### 2.4. Fermentation Profile

The fermentation profile of CNCM I-4884 was determined using API 50 CHL strips (Biomérieux, Marcy-L’Etoile, France) according to the manufacturer instructions.

### 2.5. Acid and Bile Salts Tolerance

For acid tolerance, CNCM I-4884 suspension was adjusted to 1.0 (OD 600 nm) with phosphate buffered saline (PBS) at pH 2.0 using 1 N HCl. After 30, 60, 90 and 120 min of incubation at 37 °C, suspensions were serially diluted in PBS and plated onto MRS agar. Viable cells were counted after 48 h incubation at 37 °C. For bile salts tolerance, MRS broth was supplemented with a mix of taurocholic acid, taurochenodeoxycholic acid, glycocholic acid and glycochenodeoxycholic acid (Sigma, Saint Louis, MO, USA) at 0.05 mg/mL, 0.1 mg/mL, 0.5 mg/mL, 1 mg/mL and 2 mg/mL each. Bacterial growth was monitored at 37 °C every 30 min for 24 h by OD at 600 nm using Infinite M200 Pro spectrophotometer (TECAN, Lyon, France). The experiments were performed in three biological replicates.

### 2.6. Bile Salt Hydrolase Activity

CNCM I-4884 was grown in Keiser’s modified TYI-S-33 medium (KM) adjusted to pH 6.0 and supplemented with 10% heat-inactivated fetal calf serum (FCS) for 8 h at 37 °C under anaerobic conditions. Two ml of supernatant was transferred to a new glass tube and a mixture of conjugated bile acids (BA) was added at a final concentration of 5 μg/mL each. The samples were then vortexed and kept at 37 °C. For each sample, experiments were stopped at different time points (0, 15, 30, 60 min and 2 h) using three steps: (1) addition of 2 mL of acetonitrile supplemented with internal standard (0.5 μg/mL), (2) shaking, (3) centrifugation (3500 rpm for 20 min) and (4) the upper phase was transferred to a new tube, then evaporated under nitrogen, and finally resuspended in 150 μL of methanol. Direct enzymatic activities were detected in each experiment by the decrease of conjugated BA, and the increase of unconjugated BA. BA concentrations were estimated by the area under curve (AUC) of the specific HPLC MS/MS BA analysis and reported as the mean BA in each experiment. The experiments included three biological replicates.

### 2.7. Antibiotic Susceptibility

The antibiotic susceptibility of CNCM I-4884 was tested by Biosafe (Biological Safety Solutions Ltd., Kuopio, Finland). Susceptibility to gentamicin, kanamycin, streptomycin, tetracycline, erythromycin, clindamycin, chloramphenicol, and ampicillin was tested according to ISO10932:2010 standard with VetMIC Lact-1 and VetMIC Lact-2 plates (SVA National Veterinary Institute, Uppsala, Sweden) under anaerobic conditions at 37 °C for 48 h. The minimum inhibitory concentrations (MIC) were compared with cut-off values reported by the EFSA for the obligate heterofermentative *Lactobacillus* group [32].

## 3. Results

### 3.1. Genome Sequence

A total of 15 contigs were obtained with an estimated genome length of 1,774,435 bp and an average GC content of 34.44%. Our analysis did not reveal the presence of cryptic plasmids. Genome annotation indicated the presence of 1817 protein coding sequences (CDS), 58 transfer RNA genes and 7 ribosomal RNA genes. The annotation showed 429 hypothetical proteins and 1388 proteins with functional assignments (Figure 1A). An overview of the CNCM I-4884 genome indicates that the main subsystems of this strain are protein and DNA processing, metabolism and stress response (Figure 1B).

### 3.2. Phylogenetic Position

New data obtained through the investigation of the Average Nucleotide Identity (ANI) of the whole genome sequence of CNCM I-4884 and *L. johnsonii* NCC 533 and *L. gasseri* ATCC 33323 as reference organisms using OrthoANI [39], suggest that this strain belongs to the species *L. johnsonii* and not *L. gasseri* as initially identified based on the 16S gene sequence. Indeed, according to the ANI calculation, the genome of CNCM I-4884 shares 96.79% and 85.48 % of the sequence with the genomes of *L. johnsonii* NCC 533 and *L. gasseri* ATCC 33323, with 57.38% and 54.44% overlapping nucleotide sequences, respectively. The cut-off point commonly used to separate bacterial species based on their genomic sequences is 94 to 96% ANI [36,40], suggesting that CNCM I-4884 should be reclassified as *L. johnsonii*. The phylogenetic tree based on whole genome nucleic sequence confirmed the ANI calculations, as CNCM I-4884 grouped with other *L. johnsonii* strains, namely BS15, DPC 6026, NCC533, N6.2, UMNLJ21, UMNLJ22, FI9785 and ATCC 332000 (Figure 2).

To go deeper in the molecular identification of CNCM I-4884 at the strain level, the nucleic sequences of 14 phylogenetic markers of CNCM I-4884 were compared with those of available complete genomes of *L. johnsonii* strains using BLASTn (Table 1). The results indicate that the phylogenetic markers of CNCM I-4884 share strong nucleic sequence similarity with those of *L. johnsonii* ATCC 33200 (sequence identity > 99% for all 14 phylogenetic markers tested).

The genomes of *L. johnsonii* were translated and compared with the proteins observed in the genome of strain CNCM I-4884. The comparison and visualization of the percent protein identity was generated with the SEED server version 2.0 [38]. The results confirmed that the closest strain to CNCM I-4884 is *L. johnsonii* ATCC 33200 (Figure 3).

Altogether, these results confirm that CNCM I-4884 was incorrectly classified as *L. gasseri* and instead belongs to *L. johnsonii* species complex. The closest strain to *L. johnsonii* CNCM I-4884 is *L. johnsonii* ATCC 33200. According to the ANI calculation, the genome of CNCM I-4884 shares 99.98% of sequence identity with the genome of *L. johnsonii* ATCC 33200, with 78.06% of nucleotide sequences overlapping.

### 3.3. Biosynthetic Capacities

No biosynthetic pathway was identified in the genome of CNCM I-4884 for most amino acids, including tryptophan, proline, methionine, histidine, valine, leucine, threonine, isoleucine, tyrosine and phenylalanine. Incomplete biosynthetic pathways were detected for serine, glycine, cysteine and lysine. However, CNCM I-4884 can synthesize aspartate from fumarate via aspartate ammonia-lyase (*aspA*) and alanine from cysteine via cysteine desulfurase (*iscS*). The presence of glutamine synthetase (*glnA*) and serine hydroxymethyltransferase (*glyA*) indicates that CNCM I-4884 is able to interconvert glutamine and glutamate, and serine and glycine, respectively. CNCM I-4884 could assimilate ammonia by converting L-glutamine to L-glutamate through glutamine synthetase (*glnA*) or by synthesizing L-asparagine from L-aspartate through asparagine synthetase (*asnA*). CNCM I-4884 is auxotrophic for purine and pyrimidine nucleotides. CNCM I-4884 lacks homologs for the enzymes necessary to synthesize many cofactors such as folate, thiamin, riboflavin, biotin, cobalamin, pantothenate, nicotinate, nicotinamide, cofactor A and pyridoxine.

The genome of CNCM I-4884 encodes 15 ATP binding cassette (ABC)-type transporters, which translocate both amino acids and oligopeptides. An ABC-type transporter was identified for spermidine and putrescine (*potA-D*). The CNCM I-4884 genome encodes 25 putative PTS systems (PTS) with predicted specificities for fructose, lactose, galactose, sorbose, maltose, sucrose, trehalose, mannose, tagatose, cellobiose and N-acetylglucosamine. This large number of predicted transporters was complemented by an array of 15 peptidases and proteases. To confirm the carbohydrate metabolism predictions, Analytical Profile Index (API) identification was performed using rapid API-50 CH biochemical test kit. Biochemical activity was positive for D-galactose, D-glucose, D-fructose, D-mannose, N-Acetylglucosamine, arbutine, esculine, salicine, D-cellobiose, D-maltose, D-lactose, D-saccharose, D-raffinose, amidon, gentibiose and D-tagatose (Table 2).

### 3.4. Stress Resistance

The genome of CNCM I-4884 was searched for stress-related genes. CNCM I-4884 genome encodes genes involved in cell envelope modifications, such as the *eps* gene cluster (*epsA-E*, *epsIJ*, and *epsUV*), the *dlt* operon and *cfa2*, which have been associated with improved survival under gut conditions [41,42,43,44,45]. CNCM I-4884 is also well equipped for DNA and protein protection in response to stress. DNA repair and protection genes were identified: the transcriptional regulators *lexA* and *recA*, SOS regulon *recADFJNOR*, *ruvAB*, and *ssb*, the exonuclease ATP-binding cassette complex *uvrABCD, mfd* and *xth*, the homology-independent facilitator complex *gyrAB* and *topA*, and low-fidelity DNA polymerases *dinB* and *dnaE* [46]. In order to maintain the protein integrity under stress conditions, CNCM I-4884 encodes a large collection of chaperone-active proteins such as GroES, GroEL, GrpE, HtpG, Hsp33, Hsp20, DnaK, DnaJ, HtrA, FtsH, EF-Tu, EF-G and the proteolytic Pta-AckA pathway, especially pivotal for long-term acid and bile stress resistance [46,47,48,49,50]. CNCM I-4884 also encodes the well-conserved heat shock proteins ClpA, ClpC, ClpE, ClpP, ClpX, HslU and Hslv, particularly important for the rapid response of lactobacilli when encountering adverse conditions in the gut [51,52,53,54]. CNCM I-4884 is able to cope with oxidative stress with two thioredoxin genes (*trxA* and *trxH*), thioredoxin reductase (*trxB*), glutathione reductase (*gshR1*), and two methionine sulfoxide reductase (*msrA* and *mrsB*) [55,56,57]. CNCM I-4884 encodes efflux systems for active clearance of acid- and bile-related stress factors including the MFS transporter (*mdrT*), ABC transporters (*opuA*, *oppA*, *c**cmA*, *mdlA* and *mdlB*) and Na^+^/H^+^ antiporters (*napA*, *nhaP* and *trkA*) [41,51,52,58,59,60].

The acid tolerance of CNCM I-4884 was tested at pH 2.0 (Figure 4A). CNCM I-4884 showed good acid tolerance, with more than 50% viability after 60 mn of exposure. When grown in the presence of bile salts, CNCM I-4884 is able to grow in the presence of 0.05 to 0.5 mg/mL bile salts (Figure 4B). These results suggest that the strain may not require protective encapsulation to successfully pass through the host GIT.

Furthermore, the good tolerance of CNCM I-4884 to bile salts is explained by the presence of three genes encoding bile salt hydrolases (BSH), involved in bile detoxification. The BSH activity of CNCM I-4884 was assessed for a panel of tauro- and glyco-conjugated bile salts (Figure 5). The results indicate BSH activity towards both tauro- and glyco-conjugated substrates, with a maximum activity for taurochenodeoxycholic acid (TCDCA), taurodeoxycholic acid (TDCA) and taurocholic acid (TCA).

### 3.5. In Silico Search for Adhesion Factors

In general, it is assumed that a good ability to adhere to the intestinal epithelium or the mucus layer is a desirable trait for probiotic bacteria, as this may increase the residence time in the GIT and facilitate interactions with host cells. Among the different factors involved in epithelial adhesion, exopolysaccharides (EPS) often play a role in the non-specific interactions of lactobacilli with epithelial cells. EPS may serve as a protective layer against harsh conditions during GIT transit. Once the bacteria reach the colon, EPS are involved in the interaction with the intestinal mucosa, with both positive and negative effects on the adhesion (depending on the literature) [61,62]. In general, the presence of EPS could reduce adhesion to intestinal cells, due to shielding of surface macromolecules acting as adhesins, or electrostatically interfere with the binding to receptors of mucosal surface. However, EPS could also act as ligands, mediating specific adhesion and co-aggregation. CNCM I-4884 possess nine genes of the EPS cluster (*epsA-epsE, epsI, epsJ, epsU and epsV*) but lacks homologs of EpsF. Another factor involved in adhesion to intestinal epithelial cells is lipoteichoic acid (LTA). LTAs are major components of the cell-wall of most Gram-positive bacteria and are often replaced by glycosyl or D-alanyl (D-Ala) esters by the action of four gene products (DltA to DltD). The four genes of the *dlt* operon were identified in the genome of CNCM I-4884. D-alanylation has been shown to result in enhanced adhesion to intestinal epithelial cells and colonization of the GIT in vivo [63].

The presence of four genes (*apf1*, *apf2, gtfA* and *inu*) encoding aggregation promoting factors (APF) in the genome of CNCM I-4884 is likely responsible for the strong aggregation phenotype of the strain (data not shown). APFs have been shown to contribute to survival during passage through the GIT by influencing stress tolerance and interacting with the host intestinal mucosa, as well as fibronectin, mucin and laminin, contributing to colonization [64,65,66,67,68]. In addition, the genome of CNCM I-4884 contains other genes involved in adhesion to extracellular matrix components such as fibronectin binding proteins (*fbpA, enoA1* and *enoA2*) and mucus-binding proteins (*mub, msa, mapA, srtA*) [69,70,71,72].

### 3.6. Antibiotic Resistance

The RGI program was used to comprehensively scan the genome of CNCM I-4884 against the CARD database of antibiotic resistance genes, as well as mutations in antibiotics targets known to confer resistance [73]. No known resistance genes were found. The susceptibility of CNCM I-4884 to a panel of eight antibiotics was determined by the Minimum Inhibitory Concentration (MIC) assay and compared with the most recent EFSA breakpoint values (Table 3). The MIC obtained for gentamycin, kanamycin, streptomycin, erythromycin, clindamycin, chloramphenicol and ampicillin are below the cut-off values. Although the MIC of tetracycline exceeds the cut-off concentration, this is within the inter-laboratory variation of MICs reported for non-enterococcal lactic acid bacteria [74]. Therefore, it was concluded that CNCM I-4884 was sensitive to all antibiotics of human and animal relevance requested by EFSA.

### 3.7. Virulence Factors

The VF Analyser pipeline from the Virulence Factor Database [75] and VirulenceFinder 2.0.3 of the Center for Genomic Epidemiology [76] were used to search for offensive virulence factors of major bacterial pathogen species of medical importance in the genome of CNCM I-4884. The search results in these two databases were negative. Therefore, CNCM I-4884 could be considered free of virulence factors and safe for probiotic applications.

## 4. Discussion and Conclusions

*L. johnsonii* CNCM I-4884 was selected from a previous screening based on anti-*Giardia* activity in vitro and in vivo in a murine model of giardiasis. This strain significantly antagonizes parasite growth with a drastic reduction in small intestinal trophozoite load and cyst excretion. The protective effect of CNCM I-4884 was higher to that of the reference strain *L. johnsonii* La1 [31]. The aim of this study was the identification of CNCM -4884 based on the whole genome sequence, as well as the assessment of its safety and probiotic properties.

To be considered suitable for use as a probiotic, a candidate bacterial strain must be unequivocally identified. The 16S gene sequence of strain CNCM I-4884 shares strong similarity with *L. gasseri* species, so CNCM I-4884 was originally classified as *L. gasseri*. To confirm the identity of CNCM I-4884, whole genome sequencing was performed using Illumina technology. The new results led to the reclassification of CNCM I-4884 as *L. johnsonii*, a species closely related to *L. gasseri* (which shares the same phylogenetic species complex) [36,77]. The closest strain to CNCM I-4884 is currently *L. johnsonii* ATCC 33200.

CNCM I-4884 is a strain that appears unable to synthesize *de novo* most amino acids and vitamins. Thus, this strain is dependent on large amounts of exogenous amino acids and peptides to supply protein synthesis and is therefore restricted to environments rich in such substrates. This high degree of auxotrophy is found in most lactobacilli, especially among species of the *L. acidophilus* group and reflects their demanding nutritional requirements when grown on synthetic media [13,78,79]. CNCM I-4884 counterbalances its limited biosynthetic capacities by relying on enhanced transport systems to acquire cofactors, amino acids, and other essential precursors exogenously. Its complex proteolytic system also likely provides CNCM I-4884 with a selective advantage, as acquisition of amino acids from the environment is energetically more favorable than *de novo* synthesis. CNCM I-4884 possess diverse carbohydrate transport systems and catabolic potential, which have previously been suggested to promote the ability to compete in environmental niches where sugar molecules may be abundant, such as the upper GIT of mammalian hosts [80,81].

The ability to survive the harsh conditions of the GIT is one of the vital characteristics that allows a probiotic bacterial strain to transiently colonize the host. Genome analysis of CNCM I-4884 predicts that the strain is well equipped to deal with different environmental stresses, such as acid, bile and oxidative stresses. In vitro tests revealed that CNCM I-4884 showed a good tolerance to pH 2.0 and to conjugated bile salts. In vitro BSH activity results showed good hydrolytic activity against both tauro- and glyco-conjugated BA. BA are synthesized from cholesterol and are subsequently conjugated with either taurine or glycine in the liver and finally released into the duodenum. Upon reaching the colon, BSH enzymes of the commensal microbiota modify bile salts by hydrolyzing the amide bond between amino acids and cholesterol. The role of BSHs in bile detoxification has been well demonstrated. High concentrations of bile could dissolve phospholipids and alter the structure of the lipid bilayer of membranes, causing their lysis. In addition, BA induce protein misfolding, oxidative damage to DNA and RNA, and intracellular acidification. BSH-producing bacteria can be protected by the formation of weaker unconjugated counterparts with lower solubility and emulsification capacity [82]. In addition, the liberated glycine and taurine could be used as carbon, nitrogen, and energy sources, conferring a nutritional advantage on hydrolytic bacterial strains [83]. BSH enzymes are almost exclusively associated with mammals gut-colonizing bacteria and confer a clear selective advantage in the GIT environment [84,85].

Furthermore, to exert a better beneficial effect, a probiotic strain must adhere to the intestinal mucosa or upper mucus layer. A genome-wide search of CNCM I-4884 for adhesion factors revealed the presence of a significant number of genes involved in auto-aggregation, adhesion to intestinal epithelial cells and binding to several components of the extracellular matrix, such as fibronectin, laminin and mucin. Altogether, these results suggest that CNCM I-4884 could persist in vivo by transiently colonizing the mucus layer and/or the intestinal epithelium.

The ability of a probiotic strain to transfer antibiotic resistance elements to opportunistic or pathogenic organisms through horizontal gene transfer is of paramount concern. Phenotypic testing based on MIC determination revealed that CNCM I-4884 is sensitive to most antibiotics relevant for use in humans and animals (according to the EFSA guidelines). A whole genome sequence search for the presence of known antimicrobial resistance genes confirmed the absence of acquired or transferable resistance factors and plasmids.

In conclusion, the results obtained in silico and in vitro demonstrate that *L. johnsonii* CNCM I-4884 represents a safe candidate for probiotic applications.

## Figures and Tables

**Figure 1 microorganisms-10-00273-f001:**
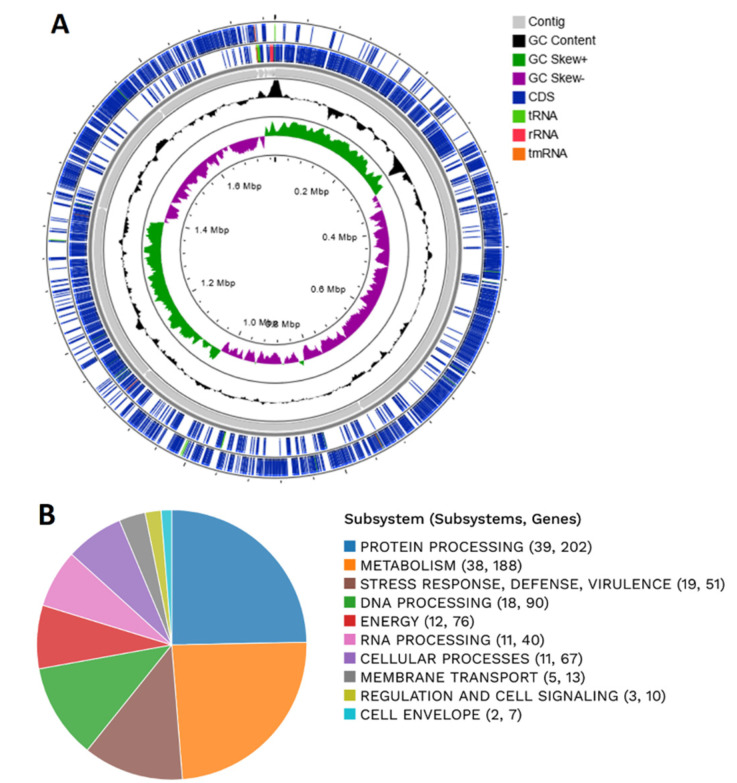
(**A**) Circular graphical map of *L. johnsonii* CNCM I-4884 genome. From outer to inner rings, CDS on the forward strand, CDS on the reverse strand, contigs, GC content and GC skew. (**B**) An overview of the RAST annotation and subsystems.

**Figure 2 microorganisms-10-00273-f002:**
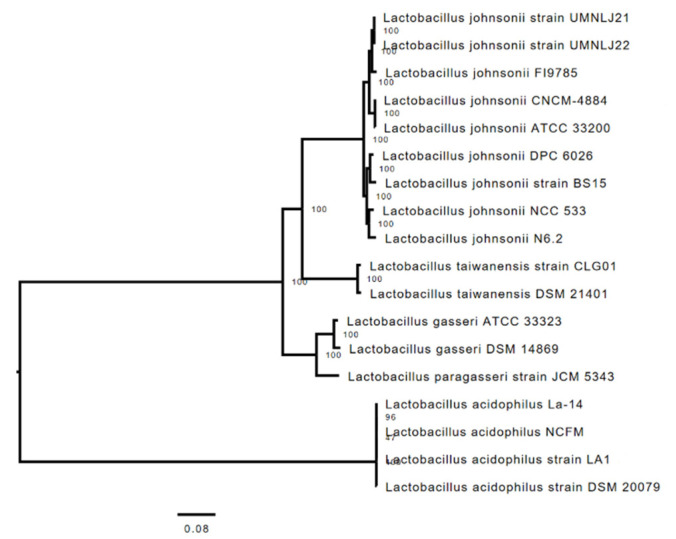
Phylogenetic relationship among selected lactobacilli genomes and strain CNCM I-4884.

**Figure 3 microorganisms-10-00273-f003:**
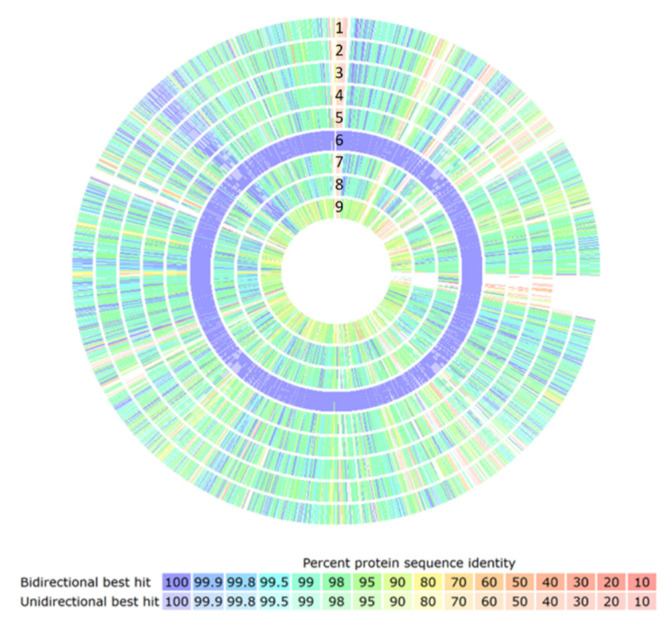
Comparison of protein identity in % of *L. johnsonii* and *L. gasseri* strains using CNCM I-4884 as reference. 1, *L. johnsonii* UMNLJ22; 2, *L. johnsonii* N6.2; 3, *L. johnsonii* FI9785; 4, *L. johnsonii* DPC 6026; 5, *L. johnsonii* BS15; 6, *L. johnsonii* ATCC 33200; 7, *L. johnsonii* NCC 533, 8, *L. johnsonii* UMNLJ21; 9, *L. gasseri* ATCC 33323. The red bar represents 10% identity and the blue bar represents 100%.

**Figure 4 microorganisms-10-00273-f004:**
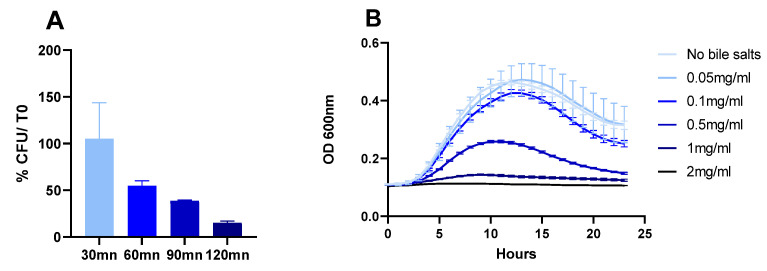
CNCM I-4884 tolerance to (**A**) pH 2.0 and to (**B**) conjugated bile salts.

**Figure 5 microorganisms-10-00273-f005:**
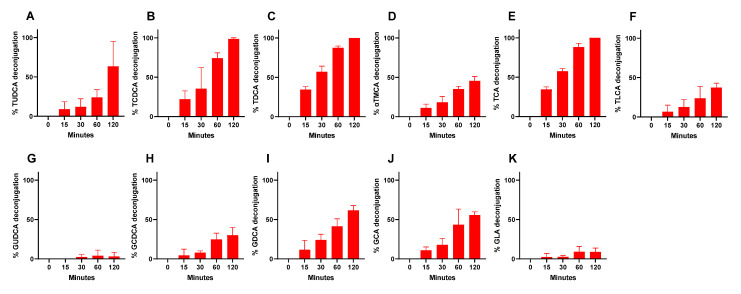
Bile acids deconjugation activity of CNCM I-4884 supernatant (**A**–**K**) TUDCA, tauroursodeoxycholic acid; TCDCA, taurochenodeoxycholic acid; TDCA, taurodeoxycholic acid; TCA, taurocholic acid; TLCA, taurolitocholic acid; αTMCA, α-tauromuricholic acid; GUDCA, glycoursodeoxycholic acid; GCDCA, glycochenodeoxycholic acid; GDCA, glycodeoxycholic acid; GCA, glycocholic acid; GLC, glycolitocholic acid. Values are expressed as percentage ± SEM (standard error of the mean). Data are the means of three independent replicates.

**Table 1 microorganisms-10-00273-t001:** Percentage of nucleic sequence identity for phylogenetic markers of CNCM I-4884 compared to homologous markers in *L. johnsonii* and *L. gasseri* strains.

Gene	*L. johnsonii* NCC 533	*L. johnsonii* DPC 6026	*L. johnsonii* UMNLJ22	*L. johnsonii* UMNLJ21	*L. johnsonii* BS15	*L. johnsonii* ATCC 33200	*L. johnsonii* N6.2	*L. johnsonii* FI9785	*L. gasseri* ATCC 33323
*Ef-tu*	99.66%	99.92%	99.92%	99.92%	99.92%	100.00%	99.92%	100.00%	98.11%
*fusA*	99.28%	99.19%	99.71%	99.71%	99.20%	100.00%	99.24%	99.71%	95.57%
*gpmA*	99.57%	99.13%	99.42%	99.42%	99.58%	100.00%	99.57%	99.42%	96.83%
*gyrA*	97.51%	97.43%	99.40%	99.40%	97.39%	100.00%	96.02%	99.28%	88.48%
*gyrB*	96.95%	96.90%	99.44%	99.44%	96.95%	100.00%	96.90%	99.39%	90.30%
*ileS*	97.97%	97.20%	99.64%	99.64%	97.12%	100.00%	97.20%	97.77%	86.33%
*lepA*	99.45%	99.41%	99.03%	99.03%	98.12%	100.00%	99.45%	99.25%	94.99%
*leuS*	98.72%	98.82%	99.59%	99.59%	98.78%	100.00%	98.79%	99.65%	89.03%
*pyrG*	98.83%	96.86%	99.69%	99.69%	96.86%	100.00%	98.58%	99.63%	89.91%
*recA*	99.44%	99.10%	99.54%	99.54%	98.98%	100.00%	98.98%	99.36%	87.22%
*recG*	96.76%	97.06%	96.57%	96.57%	97.01%	100.00%	96.18%	97.06%	85.01%
*rplB*	99.76%	99.62%	99.64%	99.64%	99.64%	100.00%	99.76%	99.64%	97.37%
*rpoB*	99.06%	99.18%	99.75%	99.75%	98.39%	99.97%	99.06%	99.67%	93.90%
*rpsC*	99.40%	99.40%	99.70%	99.70%	99.40%	100.00%	99.40%	99.55%	97.76%

**Table 2 microorganisms-10-00273-t002:** Carbohydrate fermentation profile of CNCM I-4884 assessed by API 50 CH test strips.

Substrate	Result	Substrate	Result
Glycerol	−	Esculine	+
Erythritol	−	Salicine	+
D-arabinose	−	D-cellobiose	+
L-arabinose	−	D-maltose	+
D-ribose	−	D-lactose	+
D-xylose	−	D-melibiose	−
L-xylose	−	D-saccharose	+
D-Adonitol	−	D-threalose	−
Methyl-βD-xylopyranoside	−	Inulin	−
D-galactose	+	D-melezitose	−
D-glucose	+	D-raffinose	+
D-fructose	+	Starch	+
D-mannose	+	Glycogene	−
L-sorbose	−	Xylitol	−
L-rhamnose	−	Gentiobiose	+
Dulcitol	−	D-turanose	−
Inositol	−	D-lyxose	−
D-mannitol	−	D-tagatose	+
D-sorbitol	−	D-fucose	−
Methyl-αD-mannopyranoside	−	D-arabitol	−
Methyl-αD-glucopyranoside	−	Potassium gluconate	−
N-acetylglucosamine	+	Potassium 2-cetogluconate	−
Amygdaline	−	Potassium 5-cetogluconate	−
Arbutine	+		

**Table 3 microorganisms-10-00273-t003:** Minimum Inhibitory Concentrations (MIC) of antibiotic required by EFSA for CNCM I-4884.

Antibiotic	MIC µg/mL	EFSA MIC Cut-Off µg/mL
Gentamycin	≤0.5	16
Kanamycin	8	64
Streptomycin	8	64
Tetracycline	16	8
Erythromycin	0.5	1
Clindamycin	0.12	4
Chloramphenicol	4	4
Ampicillin	1	2

## Data Availability

The whole genome sequence has been deposited at GenBank under the accession JAIQXC000000000.

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
