# Peer review of "Genome Sequence and Assessment of Safety and Potential Probiotic Traits of Lactobacillus johnsonii CNCM I-4884"

_microorganisms, 2022, doi:10.3390/microorganisms10020273_

Round 1

Reviewer 1 Report

The manuscript is well written, and the topic is interesting. However, there are some sections that need some clarification. Therefore, the manuscript must be revised before publication on "Microorganisms".

Title: the title is to be revised, as potential probiotic activities of the strain are tested in this manuscript.

I would suggest: “Genome sequencing and assessment of safety and potential probiotic traits of Lactobacillus johnsonii CNCM I-4884”

Please, be careful to write the species names in italics. Check in Table 1

Figure 5: The use of standard deviation bars is unclear. They are present for some curves and not for others; please revise the graphs and indicate the standard deviation for all points and all curves

Paragraph "3.6. Adhesion to Intestinal Epithelium": This is an important probiotic trait; however, it was not phenotypically evaluated in this study, unlike other parameters. Please, include a proper assay on adhesion capability of CNCM I-4884 (which is different from auto-aggregation ability).

Conclusion: “To conclude, the results obtained in silico and in vitro demonstrate that L. johnsonii CNCM I-4884 represents a safe candidate probiotic strain.” The conclusions are too speculative; these are preliminary data, and further tests and in vivo validations are needed to prove that the strain is truly probiotic. Please review conclusions and, as  a consequence, the Abstract.

Author Response

Reviewer 1

General Comments:

The manuscript is well written, and the topic is interesting. However, there are some sections that need some clarification. Therefore, the manuscript must be revised before publication on "Microorganisms".

Author’s answer: We appreciate your feedback. We have carefully read each of your comments and have made our answers to these. Below we explain each of the corrections that have been made to the revised version of our manuscript.

Title: the title is to be revised, as potential probiotic activities of the strain are tested in this manuscript.

I would suggest: “Genome sequencing and assessment of safety and potential probiotic traits of Lactobacillus johnsonii CNCM I-4884”..

Author’s answer: Following the advice of the reviewer we have modified the title.

Please, be careful to write the species names in italics. Check in Table 1

Author’s answer: Done.

Figure 5: The use of standard deviation bars is unclear. They are present for some curves and not for others; please revise the graphs and indicate the standard deviation for all points and all curves.

Author’s answer: We have modified this graph and now the standard deviation are clearer.

Paragraph "3.6. Adhesion to Intestinal Epithelium": This is an important probiotic trait; however, it was not phenotypically evaluated in this study, unlike other parameters. Please, include a proper assay on adhesion capability of CNCM I-4884 (which is different from auto-aggregation ability).

Author’s answer: Indeed, in this work, the adhesion capacity of the CNCM I-4884 strain to different intestinal cell lines was evaluated in vitro. However, the adhesion results could not be included due to an ongoing patent application. Therefore, we have decided to change the title of this section as follows: "In silico search for adhesion factors".

Conclusion: “To conclude, the results obtained in silico and in vitro demonstrate that L. johnsonii CNCM I-4884 represents a safe candidate probiotic strain.” The conclusions are too speculative; these are preliminary data, and further tests and in vivo validations are needed to prove that the strain is truly probiotic. Please review conclusions and, as a consequence, the Abstract.

Author’s answer: Please note that the probiotic properties of the CNCM I-4884 strain have been previously described in a publication of our team (ref: Allain T, Chaouch S, Thomas M, Travers MA, Valle I, Langella P, Grellier P, Polack B, Florent I, Bermúdez-Humarán LG. Bile Salt Hydrolase Activities: A Novel Target to Screen Anti-Giardia Lactobacilli? Front Microbiol. 2018 Feb 8;9:89). Indeed, in this former publication we have clearly demonstrated that strain CNCM I-4884 possesses anti-Giardia activity both in vitro and in vivo in a murine model of giardiasis.

Also, the abstract and discussion have been modified to add these previous observations.

Reviewer 2 Report

In this study, the authors have studied one strain, which had exhibited tolerance to low pH and bile salts with strong bile salt hydrolase (BSH) activity. They also sequence the genome of this strain for further investigation. Some points need to be considered and corrected before considering for publication. First of all, the scientific name of all lactobacilli has been changed. The authors should consider all the new names of lactobacilli. The genus of Lactobacillus has been divided into 25 new genera. This should be regarded throughout the manuscript, including all the figures and their legends.

Line 67, how was this microbe isolated? Provide more information.

Line 72, it is not clear why the authors did TEM? Please provide the logic for this work.

Line 92, the accession number does not work. I tried, and nothing came up.

Line 97, how did you end up with these 18 strains? Why not some others? What were the criteria of selection? If it was a PATRIC automatic selection/analysis, why did you mention those genomes were downloaded from NCBI?

Line 113, Why these strange pH values (6-7) and duration (24h)? Usually, the strain survivability and tolerance are checked in low pH such as 2 or 2.5 pH for a few hours and not growth for 24h. Meanwhile, how could you keep the pH constant in MRS? The strain constantly produces lactic acid and lowers the pH during its growth. Did you consider this point?

Line 144, this is obvious information and does not need a TEM analysis since you have the taxonomy by the genome. Still, I wonder why they have performed such analysis? It needs to be discussed better.

Figure 2, why the contigs started from the biggest toward the smallest. This can not be the genome map since the contigs are in the wrong direction. It is necessary to make them in order based on the reference genome.

Line 193, all the scientific names should be italic.

Line 281, what about any phenotypical test to confirm this information? Have you proved their functionality?

Table 3, all lactobacilli are intrinsically resistant to kanamycin. However, the author reported a very low MIC value in this study. It needs to be rechecked if the value is correct.

Line 319, the author should reference this sentence that ‘a dilution step above the 318 cut-off value is acceptable’

Line 319, What about the antibiotic resistance genes reported in Figure two? The analysis by PATRIC indicated resistant genes. How do the authors interpret the contradiction?

All the references should be corrected, and all the scientific names should be italic. I see many errors.

Author Response

General Comments:
In this study, the authors have studied one strain, which had exhibited tolerance to low pH and bile salts
with strong bile salt hydrolase (BSH) activity. They also sequence the genome of this strain for further
investigation. Some points need to be considered and corrected before considering for publication. First
of all, the scientific name of all lactobacilli has been changed. The authors should consider all the new
names of lactobacilli. The genus of Lactobacillus has been divided into 25 new genera. This should be
regarded throughout the manuscript, including all the figures and their legends.
Author’s answer: According to the reclassification of the genus Lactobacillus by Zheng and colleagues
(ref: Zheng J, Wittouck S, Salvetti E, Franz CMAP, Harris HMB, Mattarelli P, O'Toole PW, Pot B,
Vandamme P, Walter J, Watanabe K, Wuyts S, Felis GE, Gänzle MG, Lebeer S. A taxonomic note on
the genus Lactobacillus: Description of 23 novel genera, emended description of the genus Lactobacillus
Beijerinck 1901, and union of Lactobacillaceae and Leuconostocaceae. Int J Syst Evol Microbiol. 2020
Apr;70(4):2782-2858), all species cited in this article (L. johnsonii, L. gasseri, L. paragasseri, L.
acidophilus and L. taiwanensis) belong to the Lactobacillus delbrueckii group, which are still named
Lactobacillus.
Line 67, how was this microbe isolated? Provide more information.
Author’s answer: L. johnsonii CNCM I-4884 was isolated from a human sample, please refer to our
previous publication for more details: Ref: Allain T, Chaouch S, Thomas M, Travers MA, Valle I,
Langella P, Grellier P, Polack B, Florent I, Bermúdez-Humarán LG. Bile Salt Hydrolase Activities: A
Novel Target to Screen Anti-Giardia Lactobacilli? Front Microbiol. 2018 Feb 8;9:89.
Line 72, it is not clear why the authors did TEM? Please provide the logic for this work.
Author’s answer: We agree with this remark and therefore these results have been withdrawn from the
revised version of our Ms.
Line 92, the accession number does not work. I tried, and nothing came up.
Author’s answer: The complete genome sequence of strain CNCM I-4884 has been deposited in the
NCBI database under the BioSample n° SAMN21406638 and Accession n° JAIQXC000000000.
However, access to the genome is still confidential and will be released once the article is published.
Line 97, how did you end up with these 18 strains? Why not some others? What were the criteria
of selection? If it was a PATRIC automatic selection/analysis, why did you mention those genomes
were downloaded from NCBI?
Author’s answer: These 18 strains were selected for their close phylogenetic position with CNCM I-
4884 and for the good quality of their whole genome sequence. The tree was generated using PATRIC
server. We used the genomes downloaded from NCBI for protein comparison analysis using the SEED
server.
Line 113, Why these strange pH values (6-7) and duration (24h)? Usually, the strain survivability
and tolerance are checked in low pH such as 2 or 2.5 pH for a few hours and not growth for 24h.
Meanwhile, how could you keep the pH constant in MRS? The strain constantly produces lactic
acid and lowers the pH during its growth. Did you consider this point?
Author’s answer: As suggested by the reviewer, we modified the design of this experiment to test the
tolerance of the strain to pH 2.0 for 1, 2 and 3 hours. For this we followed the protocol of a previous
publication of our team: Ref. Torres-Maravilla E, Lenoir M, Mayorga-Reyes L, Allain T, Sokol H,
Langella P, Sánchez-Pardo ME, Bermúdez-Humarán LG. Identification of novel anti-inflammatory
probiotic strains isolated from pulque. Appl Microbiol Biotechnol. 2016 Jan;100(1):385-396.
Line 144, this is obvious information and does not need a TEM analysis since you have the
taxonomy by the genome. Still, I wonder why they have performed such analysis? It needs to be
discussed better.
Author’s answer: This part has been withdrawn.
Figure 2, why the contigs started from the biggest toward the smallest. This can not be the genome
map since the contigs are in the wrong direction. It is necessary to make them in order based on
the reference genome.
Author’s answer: Message sent to the reviewer on January 7 via the platform:
“Dear reviewer, we are preparing our answers to your comments; however, we have one doubt about
this comment: We don't understand this question, normally this is the way PATRIC generates the genome
figures we input and also this is the way we have found in the bibliography the results are interpreted,
could you please be more explicit about your comment?”
Line 193, all the scientific names should be italic.
Author’s answer: Done.
Line 281, what about any phenotypical test to confirm this information? Have you proved their
functionality?
Author’s answer: Indeed, in this work, the adhesion capacity of the CNCM I-4884 strain to different
intestinal cell lines was evaluated in vitro. However, the adhesion results could not be included due to
an ongoing patent application. Therefore, we have decided to change the title of this section as follows:
"In silico search for adhesion factors".
Table 3, all lactobacilli are intrinsically resistant to kanamycin. However, the author reported a very low
MIC value in this study. It needs to be rechecked if the value is correct.
Author’s answer: Antibiotic resistance of the strain was evaluated by Biosafe-Biological Safety
Solutions Ltd according to ISO10932:2010. The MIC assay was performed in duplicate. Lactobacillus
paracasei strain LMG12586 (ATCC 334) was used as a quality control. This strain then served as a
positive control for kanamycin resistance (see table below).
In addition, the antibiotic resistance of CNCM I-4884 was also assessed in our laboratory using Mueller-
Hinton medium. The results showed a MIC of 32 μg/ml, which is higher than that obtained by Biosafe.
However, the strain is still considered sensitive to this antibiotic according to the EFSA cut-off values
for the obligate heterofermentative Lactobacillus group.
Line 319, the author should reference this sentence that ‘a dilution step above the 318 cut-off value is
acceptable’
Author’s answer: We have corrected this.
Line 319, What about the antibiotic resistance genes reported in Figure two? The analysis by PATRIC
indicated resistant genes. How do the authors interpret the contradiction?
Author’s answer: PATRIC analysis predicted the regions in the genome associated with antimicrobial
resistance. However, the identified genes confer susceptibility and not antimicrobial resistance. In fact,
most of them are essential genes considered as antibiotic targets, such as Alr, EF-G, EF-Tu, gyrA, gyrB,
rpoB and rpoC. Some could confer antibiotic resistance when mutated, for example folA, murA, gidB
and pgsA. None of these described mutations are present in CNCM I-4884 genome. In addition, some
of the genes identified by PATRIC are responsible for antibiotic susceptibility, for example Ddl.
All the references should be corrected, and all the scientific names should be italic. I see many errors.
Author’s answer: Done.

Dear reviewer, we are preparing our answers to your comments; however, we have one doubt about this comment:

"Figure 2, why the contigs started from the biggest toward the smallest. This can not be the genome map since the contigs are in the wrong direction. It is necessary to make them in order based on the reference genome."

We don't understand this question, normally this is the way PATRIC generates the genome figures we input and also this is the way we have found in the bibliography the results are interpreted, could you please be more explicit about your comment?

Thanks in advance

Round 2

Reviewer 1 Report

The manuscript has been revised and ameliorated.

However, (as pointed out also by reviewer 2) the BioSample n° SAMN21406638 and Accession n° JAIQXC000000000 of WGS do not work; the above numbers, in fact, are not found in NCBI database; please re-check and add the proper BioSample and Accession numbers in the manuscript (this is need before publication).  

Author Response

We have requested the release of access to the database sequence: it is available since this morning.

Reviewer 2 Report

The contigs are not representing the genome map. The purpose of showing this kind of figure is to present the correct genome map and not only a figure. It is necessary to put the contigs in order based on the reference genome. Just download the complete genome map of reference strain within the taxonomy and try to align the genome with reference. 

Author Response

We have modified Figure 1 as asked by the reviewer.